# Phenotypic and Genetic Characterization of *Klebsiella pneumoniae* Isolates from Wild Animals in Central Italy

**DOI:** 10.3390/ani12111347

**Published:** 2022-05-25

**Authors:** Alexandra Chiaverini, Alessandra Cornacchia, Gabriella Centorotola, Elga Ersilia Tieri, Nadia Sulli, Ilaria Del Matto, Giorgio Iannitto, Domenico Petrone, Antonio Petrini, Francesco Pomilio

**Affiliations:** Istituto Zooprofilattico Sperimentale dell’Abruzzo e del Molise “G. Caporale”, via Campo Boario, 64100 Teramo, Italy; a.chiaverini@izs.it (A.C.); g.centorotola@izs.it (G.C.); e.tieri@izs.it (E.E.T.); n.sulli@izs.it (N.S.); i.delmatto@izs.it (I.D.M.); g.iannitto@izs.it (G.I.); d.petrone@izs.it (D.P.); a.petrini@izs.it (A.P.); f.pomilio@izs.it (F.P.)

**Keywords:** *K. pneumoniae*, wildlife, antimicrobial resistance, whole genome sequencing

## Abstract

**Simple Summary:**

*Klebsiella pneumoniae* is an opportunistic bacterium known as a nosocomial pathogen, but it is able to colonize a multitude of ecological niches, both free-living and host-associated. For this reason, the presence of antimicrobial resistance genes in wildlife is an indicator that resistant bacteria of human- or livestock-origin are widespread in the environment. Wild animals can act as efficient reservoirs and epidemiological links between humans, livestock, and natural environments. Despite the renewal of interest in epidemiology, there is a critical lack of an unbiased ecological approaches to defining reservoirs and sources of infections. The aims of the present study were to investigate the occurrence of *K. pneumoniae* in wildlife, to characterize antimicrobial features through phenotypic and whole genome sequencing approaches, and to identify similar available sequences that could suggest a transmission event.

**Abstract:**

Despite *Klebsiella pneumoniae* being widely recognized as a nosocomial pathogen, there is a critical lack in defining its reservoirs and sources of infections. Most studies on risk factors have focused on multidrug-resistant (MDR) isolates and clinically-oriented questions. Over a two-year period, we sampled 131 wild animals including mammal and bird species from three regions of Central Italy. All typical colonies isolated from the analytical portions were confirmed by real-time PCR and identified by MALDI-TOF mass spectrometry (MALDI-TOF MS). All confirmed *K. pneumoniae* isolates were tested for antimicrobial susceptibility to 29 antimicrobials and subjected to whole genome sequencing. Typical colonies were detected in 17 samples (13%), which were identified as *K. pneumoniae* (*n* = 16) and as *K. quasipneumoniae* (*n* = 1) by MALDI-TOF MS. The antimicrobial susceptibility profile showed that all the isolates were resistant to β-lactams (ceftobiprole, cloxacillin, cefazolin) and tetracycline; resistance to ertapenem and trimethoprim was observed and nine out of 16 *K. pneumoniae* isolates (56.2%) were classified as MDR. Genomic characterization allowed the detection of fluoroquinolone resistance-associated efflux pumps, fosfomycin and β-lactamase resistance genes, and virulence genes in the overall dataset. The cluster analysis of two isolates detected from wild boar with available clinical genomes showed the closest similarity. This study highlights the link between humans, domestic animals, and wildlife, showing that the current knowledge on this ecological context is lacking and that the potential health risks are underestimated.

## 1. Introduction

The Central Italy region is characterized by a considerable extent of rural areas and natural parks, which host many wildlife mammal and birds’ species, and at the same time by the presence of urbanized and industrial areas. The overlapping of the human activities and natural habits of the wildlife can increase the spread of resistant bacteria between different ecological niches [1]. Wild animals could act as efficient sentinel and antimicrobial resistance (AMR) reservoirs for this spread, representing an epidemiological link between humans, livestock, and natural environments.

In this scenario, it is of utmost importance to examine AMR through a “One Health” perspective [1]. This perspective contemplates an integrated and holistic multidisciplinary approach, highlighting the importance of the better integration of human, livestock, wildlife, and environmental aspects, in order to identify key priorities for tackling AMR [2].

Although several authors have described the occurrence of AMR bacteria, such as *Enterobacteriaceae*, in wild animals [3,4,5,6,7], little data and evidence are available about the role played by wild fauna in the maintenance and transmission of *Klebsiella pneumoniae* (*K. pneumoniae)* in the environment, especially in the Italian context [8,9,10,11].

*Klebsiella pneumoniae* is an opportunistic pathogen belonging to the ESKAPE group, which includes five other nosocomial pathogens that exhibit multidrug resistance and virulence: *Enterococcus faecium*, *Staphylococcus aureus*, *Acinetobacter baumannii*, *Pseudomonas aeruginosa*, and *Enterobacter* spp. [12]. *K. pneumoniae* is a prominent cause of nearly 10% of nosocomial infections in Western countries [13] and is known for high phenotypic and genetic diversity, in particular regarding antimicrobial-resistance genes (ARGs) and plasmid burden [14]. It evades antimicrobial action with a variety of mechanisms including: enzymatic degradation or inactivation of antimicrobial compounds, changing of membrane permeability, and modifying the target site of antimicrobial compounds by mutation of bacterial proteins [15]. All these mechanisms induce the development of multidrug-resistant (MDR) strains that are characterized by resistance to more than three classes of drugs in addition to ampicillin [16].

Although this pathogen is known for its wide ecological distribution and has been detected in industrial wastewaters, saltwater and freshwater, plant products, fresh vegetables, living trees, and plants and plant by-products [14], most of the studies that investigate the interface between humans, animals, and the environment focus only on companion and livestock animals, and on extended spectrum β-lactam-resistant *K. pneumoniae* [15].

For these reasons, the aim of the present study was to investigate the presence of *K. pneumoniae* in a still unknown ecological niche, that of wild animals in Central Italy, through the analysis of phenotypic antimicrobial resistance characteristics and data derived from whole genome sequencing (WGS). In addition, identification and matching with similar available sequences has been performed to highlight the presence of a transmission event.

## 2. Materials and Methods

### 2.1. The sample Collection

During the period 2020–2021, 119 wild animals including 10 mammal species (wild boar *(Sus scrofa*), red deer (*Cervus elaphus*), roe deer (*Capreolus capreolus*), fallow deer *(Dama dama*)*,* European badger (*Meles meles*)*,* red fox (*Vulpes vulpes*)*,* wolf (*Canis lupus italicus*), porcupine (*Erethizon dorsatum*)*,* otter (*Lutra lutra*), and hedgehog (*Erinaceus europaeus*)) and 5 bird species (magpie (*Pica pica*), jay (*Garrulus glandarius*), goose (*Anser anser*), sparrow hawk (*Accipiter nisus*) and starling (*Sturnus vulgaris*)) were collected in Central Italy, mainly in Abruzzo, Molise, and Lazio Regions. Most of them were collected as part of the regional plans of epidemiological surveillance and monitoring of diseases in wild fauna. Bird species were collected in the frame of the national plan of surveillance of West Nile disease and Usutu. Wild animals of this study died due to anthropic activities (gunshot wounds or impacts with vehicles) and the carcasses were subjected to necropsy for feces and tissue sample collection. A total of 131 diagnostic samples including faces (*n* = 98), intestine (*n* = 13), and brain (*n* = 20) were collected and analyzed at Istituto Zooprofilattico Sperimentale dell’Abruzzo e del Molise “G. Caporale”.

### 2.2. Bacterial Isolation and Identification

Analytical portions of 25 g of each sample were enriched in 225 mL of Buffered peptone water (BPW) (Biolife Italiana, Monza, Italy) for 24 ± 1 h at 37 ± 1 °C. For analytical portions lower than 25 g, a volume of BPW to maintain a weight/volume ratio of 1:10 was added to the sample. Thereafter, the enrichment was streaked onto Simmons Citrate agar + Inositol (SCAI) and onto *Klebsiella* ChromoSelect Selective agar base + selective supplement (Sigma-Aldrich, St. Louis, MO, USA), followed by incubation at 44 ± 1 °C for 48 ± 1 h.

Both from SCAI and *Klebsiella* ChromoSelect Selective agar, typical yellow mucoid colonies and purple colonies, respectively, were picked up and subcultured onto nutrient agar (Microbiol & C., Cagliari, Italy) at 37 ± 1 °C for 24 ± 1 h, in order to confirm them by real-time PCR.

### 2.3. Real-Time PCR

All typical colonies were subjected to DNA extraction using DNeasy Blood and Tissue Kit (Qiagen, Hilden, Germany). Then a real-time PCR was performed in order to detect *K. pneumoniae* and related species (Kp1 to Kp7 phylogroups, making up the Kp complex) by specific amplification of the zikr region according to Barbier et al. [17] with minor modifications. All real-time PCR assays were performed on a 7500 Fast Real-Time PCR System (Thermo Fisher Scientific, Lenexa, KS, USA) with the following temperature program: 95 °C for 20 s and 40 cycles at 95 °C for 3 s and 60 °C for 30 s. Melt curves were generated with temperature increments of 0.3 °C per cycle from 60 to 95 °C. DNA was amplified in a 20 µL PCR mix containing 10 µL of Kapa syber fast rox qPCR MasterMix (Roche, Basel, Switzerland), 2 µL of each primer (final concentration 300 nM), 2.5 µL of template DNA, and 3.5 µL PCR grade water (Thermo Fisher Scientific).

### 2.4. MALDI-TOF Mass Spectrometry

All real time PCR positive colonies were subjected to matrix-assisted laser desorption/ionization time-of-flight mass spectrometry (MALDI-TOF MS) analysis.

In brief, bacterial pure and fresh colonies were applied in a 96-well steel target plate (Bruker Daltonik GmbH, Bremen, Germany) using a toothpick overlaid with 1 μL of α-cyano-4-hydroxycynnamic acid–matrix solution (HCCA, Bruker Daltonik GmbH, Bremen, Germany), and air dried at room temperature (18–27 °C). For each isolate, three replicates were performed.

The acquisition and analysis of mass spectra were carried out by Microflex LT/SH™ spectrometer (Bruker Daltonik GmbH, Bremen, Germany), which was operated in linear positive mode covering the molecular weight range of 2 to 20 kDa. Each spot was hit with 240 shots in several points with a pulsed nitrogen laser (ʎ = 337 nm) at a frequency of 60 Hz. Prior to analysis, the instrument was calibrated using an extract of the *Escherichia coli* DH5α strain, with two additional proteins: RNase A of 13,683.2 Da and Myoglobin of 16,952.3 Da (BTS, Bruker Daltonik GmbH, Bremen, Germany). The mass spectra were generated automatically by the MBT Compass Explorer 4.1.70 software, which compares the sample mass spectrum to the reference mass spectra in the database and calculates a logarithmic score value (0–3.0) reflecting the similarity between the sample and the reference spectrum. If the log (score) value is ≥2.0, the bacterial identification is reliable at the species level; if it is between 1.7 and 1.99, it is reliable at the genus level, while if the log (score) is <1.7, bacterial identification is not reliable by this technique.

### 2.5. Antimicrobial Susceptibility Testing

All confirmed *K. pneumoniae* isolates were tested for antimicrobial susceptibility to 29 antimicrobials using the agar disk diffusion and the microdilution methods. The antimicrobials tested are representative of the major classes of antimicrobial drugs β-lactams, quinolones, and aminoglycosides important to both veterinary and human medicine.

For antimicrobial susceptibility testing by microdilution method, *K. pneumoniae* isolates stored in cryobank tubes (Pro-Lab Diagnostics, Richmond Hill, Canada), were streaked on nutritive agar plates (Microbiol, Cagliari, Italy) and incubated at 37 °C for 24 h. At the end of the incubation, the purity of the culture was visually checked for morphology and 3–5 colonies were picked and suspended in a tube containing 5 mL of Sensititre demineralized water (Thermo Fisher Scientific, Lenexa, KS, USA) to reach a final turbidity equal to 0.5 McFarland standard (Biomeriux Italia, Florence, Italy). Immediately after, 10 µL of each inoculum was further added to a tube containing Sensititre Cation Adjusted Mueller–Hinton Broth (SCAMHB) (Thermo Fisher Scientific, Lenexa, KS, USA).

The inoculum suspension (50 µL) in SCAMHB was dispensed to a U-bottom 96-well microtiter GN3F plate (Thermo Fisher Scientific, Lenexa, KS) containing the dried serially-diluted antimicrobials. The antibiotics in Sensititre GN3F plates were as follows: amikacin (AMI), ampicillin (AMP), ampicillin-sulbactam (A/S2), aztreonam (AZT), cefazolin (FAZ), cefepime (FEP), cephalotin (CEP), meropenem (MERO), ertapenem (ETP), cefuroxime (FUR), gentamicin (GEN), ciprofloxacin (CIP), piperacillin-tazobactam constant 4 (P/T4), cefoxitin (FOX), trimethoprim-sulfamethoxazole (SXT), cefpodoxime (POD), ceftazidime (TAZ), tobramycin (TOB), tigecycline (TGC), ticarcillin-clavulanic acid constant 2 (TIM2), ceftriaxone (AXO), and tetracycline (TET). After incubation at 37 °C for 20–24 h, the minimal inhibitory concentration (MIC) values were read using the Sensititre OptiRead Automated Fluorometric Plate Reading System (Thermo Fisher. Lenexa, KS, USA). The interpretation of MIC values was carried out in accordance with EUCAST guidelines [18], applying CLSI breakpoints [19] where no EUCAST breakpoints were available.

For antimicrobial susceptibility testing by disk diffusion method, *K. pneumoniae* isolates stored in cryobank tubes (Pro-Lab Diagnostics, Richmond Hill, Canada), were streaked on nutritive agar plates (Microbiol, Cagliari, Italy) and incubated at 37 °C for 24 h. At the end of the incubation, the purity of the culture was visually checked for morphology and 3–5 colonies were picked and suspended in a tube containing 5 mL of meat broth (IZSAM, Italy) to reach a final turbidity equal to 0.5 McFarland standard (Biomeriux Italia, Florence, Italy). Immediately after, each inoculum was spread on Mueller–Hinton agar (Liofilchem, Teramo, Italy) using a sterile swab. The antibiotics (Oxoid Ltd., Basingstoke, UK) tested by disk diffusion method were as follows: cefpodoxime-clavulanic acid (CD 11 µg), ceftobiprole (BPR 5 µg), kanamycin (KAN 30 µg), nalidixic acid (NAL), netilmicin (NET 10 µg), trimethoprim (TMP 5 µg), and cloxacillin (CLO 30 µg). The interpretation of disk diffusion results was carried out in accordance with EUCAST guidelines [18].

Multidrug-resistant (MDR) isolates were defined as those isolates resistant to ≥3 antimicrobial classes in addition to ampicillin, to which all *K. pneumoniae* isolates are known to be resistant [16].

### 2.6. Next-Generation Sequencing (NGS) and Data Analysis

All the isolates identified by MALDI-TOF MS as *K. pneumoniae* were also subjected to DNA extraction according to Portmann et al. [20]. Cultures were grown overnight in nutrient agar (Microbiol & C., Cagliari, Italy) at 37 ± 1 °C and DNA extraction was performed on single colonies using the DNeasy Blood and Tissue Kit (Qiagen, Hilden, Germany) according to the manufacturer’s instructions.

Then, 1 ng of DNA of each sample was used for library preparation using Nextera XT DNA chemistry (Illumina, San Diego, CA, USA) according to the manufacturer’s protocols. Whole genome sequencing was performed using the NextSeq 500 platform (Illumina, San Diego, CA, USA) with the NextSeq 500/550 mid output reagent cartridge v2 (300 cycles, standard 150-bp paired-end reads)

For the WGS data analysis, an in-house pipeline [21] was used, which included steps for trimming using Trimmomatic v.0.36 [22] and a quality control check of the reads with FastQC v.0.11.5 [23]. Genome de novo assembly of the reads was performed using SPAdes v.3.11.1 [24]. The genome assembly quality check was performed with QUAST [25].

### 2.7. Typing, AMR, and Virulence Gene Profiles

The species detection and capsule (K) and O antigen (LPS) serotype prediction for each draft genome were performed using Kleborate [26] and Kaptive [27], respectively. Then, the assemblies were uploaded onto the BIGSdb platform [28] to calculate the sequence type (ST) using the *K. pneumoniae* MLST scheme [29].

The presence of β-lactam-, tetracycline-, sulphonamide-, and aminoglycoside-resistant genes were verified querying the Kleborate database hosted on the Pathogenwatch platform [30]. Furthermore, these results were compared with those of ABRicate v.8.10 [31] by searching automatically known AMR and virulence predictor genes using the BLAST+ algorithm and public databases as vfdb [32], ResFinder [33], CARD [34], and ARG-ANNOT [35].

All the assemblies obtained in this study were analyzed using Kleborate to detect the presence of acquired virulence loci associated with invasive infections such as siderophores yersiniabactin (*ybt*), aerobactin (*iuc*) and salmochelin (*iro*), the genotoxin colibactin (*clb*), and the hypermucoidy locus *rmpADC*.

PlasmidFinder 2.1 [36] was used to verify the presence of plasmid replicons carrying AMR gene resistance.

Finally, the visualization of the phenotypic antimicrobial susceptibility profiles was assessed by heatmap, using the Morpheus tool [37].

### 2.8. Clustering Analysis

cgMLST analysis was performed to verify the relatedness among the 16 *K. pneumoniae* draft assemblies according to the Pasteur Institute’s scheme of 629 loci [38] hosted in BIGSdb database. The resulting neighbor joining (NJ) tree was visualized using Interactive Tree of Life (iTOL) [39].

In order to identify similar available sequences that could be indicative of a transmission event or outbreak using an allele-based method, the *K. pneumoniae* genomes that showed particular AMR and virulence patterns were uploaded onto the Pathogenwatch platform.

## 3. Results

### 3.1. Bacterial Detection and Identification Results

From a total of 131 samples tested, *K. pneumoniae* typical colonies were detected by plate culture method in 17 samples (13%). All 17 presumptive colonies were confirmed by real-time PCR and identified as *K. pneumoniae* by MALDI-TOF MS. The isolates were isolated from wild animals distributed mainly in the Abruzzo, Molise, and Lazio regions (Figure 1). The metadata related to the 17 isolates are summarized in Table 1.

### 3.2. Whole Genome Sequencing and Typing Results

The average reads quality score after trimming returned 34.1 (min 33.73, max 34.27). The mean length of the 17 draft assemblies with SPAdes was 5,375,060 (min 5,189,089; max 5,608,041) with an average number of contigs of 92.3 (min 52; max 224).

The species detection results from Kleborate, countering those obtained from MALDI-TOF, showed that 16 genomes belonged to *K. pneumoniae* and one to *K. quasipneumoniae* (2021.TE.10), isolated from a European badger. This result was not surprising since *K. quasipneumoniae* spectra were not available in the MBT Compass Library.

Among the sixteen *K. pneumoniae* genomes, Kleborate identified 11 KL and 7 O antigen serotypes. Meanwhile the single *K. quasipneumoniae* isolate was predicted as a KL35 and O3/O3a serotype.

The MLST analysis showed that the 16 *K. pneumoniae* genomes belonged to 14 different STs (23, 35, 116, 133, 162, 200, 219, 277, 301, 2217, and 3071) including three novel ones, which were coded as *5670, *ca55, and *fc60 (respectively 2021.TE.5, 2021.TE.7, and 2021.TE.15); the *K. quasipneumoniae* isolate belonged to ST4895.

All the results on genomes’ typing are reported in Table 2.

### 3.3. Antimicrobial Susceptibility Testing

The antimicrobial susceptibility profiles of isolates are reported in Figure 2. All 17 isolates (100%) were resistant at least to one antimicrobial. Regarding β-lactams, all *K. pneumoniae* isolates were resistant to ampicillin, cloxacillin, and cefazolin; on the other hand, all *K. pneumoniae* isolates were susceptible to cefpodoxime, ampicillin/sulbactam, ceftazidime, ceftriaxone, and cefuroxime. For aztreonam, ticarcillin-clavulanic acid, piperacillin-tazobactam, and cefepime, all isolates showed an intermediate susceptibility profile. Meanwhile, 3 of the 16 isolates (18.8%) were resistant to cefoxitin and 13 of the 16 isolates (81.3%) were resistant to ceftobiprole. About carbapenems, all isolates were susceptible to meropenem, but 7 of the 16 isolates (43.8%) were resistant to ertapenem. All the 16 *K. pneumoniae* isolates were susceptible to cefpodoxime-clavulanic acid, except one (2021.TE.3) that had intermediate susceptibility.

All the 16 *K. pneumoniae* isolates were susceptible to aminoglycosides (amikacin, gentamicin, netilmicin, and kanamycin), tigecycline, and quinolones (nalidixic acid). Importantly, all isolates were resistant to tetracycline and intermediate for tobramycin and ciprofloxacin. The interpretation of antimicrobial susceptibility profile was not possible for cephalotin due to the lack of both EUCAST and CLSI breakpoints.

All *K. pneumoniae* isolates were susceptible to trimethoprim-sulfamethoxazole, but 3 of 16 isolates (18.8%) were resistant to trimethoprim. As a whole, 9 of 16 isolates (56.2%) were classified as MDR. 

Finally, only the *K. quasipneumoniae* isolate (2021.TE.10) showed resistance to β-lactams (ampicillin, cefoxitin, ceftobiprole, cloxacillin and cefazolin), carbapenem (ertapenem), and tetracycline.

### 3.4. AMR and Virulence Gene Profiles Results

Regarding the AMR genetic determinants (Table 2), all the 17 isolates harbored fluoroquinolone resistance-associated efflux pumps *oqxAB*, which are core genes in *K. pneumoniae*, fosfomycin resistance (*fosA*, also core gene), and β-lactamase genes. These latter included *bla*_SHV-1_, *bla*_SHV-11_, (both core genes too) and the acquired *bla*_SHV-27_, *bla*_SHV-33_, *bla*_SHV-65_, *bla*_SHV-75_, and *bla*_OKP-A*-2*_. Additionally, the OmpK37 gene was detected in all the draft assemblies.

Tetracycline (*tetA*), sulfonamide (*sul2*), and aminoglycoside (*strAB*) resistance genes were carried only by 2021.TE.18, which was isolated from a wild boar.

Kleborate results showed different virulence loci identified as yersiniabactin (*ybt*), aerobactin (*iuc*), salmochelin (*iro*), colibactin (*clb*), and hypermucoidy regulator gene *rmpA*. Specifically, four different lineages of *ybt* (*ybt1*, *ybt5*, *ybt9*, and *ybt16*) mobilized by different integrative conjugative elements (ICEKp 1, 3, 6, and 12) were carried in six out of 17 isolates. Further, seven *K. pneumoniae* isolates harbored three different aerobactin siderophores: iuc1 found in one isolate (2021.TE.17), *iuc3* in five isolates, and one unknown iuc was detected in one isolate isolated from a wild boar (2021.TE.2). Colibactin (*clb2*) and salmochelin (*iro1*) siderophores and hypermucoid genes (rmpA1 and rmpA2) were carried by only one isolate found in a wild boar (2021.TE.17).

Lastly, PlasmidFinder results showed that nine *K. pneumoniae* isolates and the only *K. quasipneumoniae* isolate harbored the incompatibility group (Inc) plasmids (IncFIB(K), IncFIB(AP001918), IncFIB(K)(pCAV1099-114), and IncFIB(pKPHS1)). The colicin (Col)-type plasmids (Col(pHAD28)) were detected in one isolate isolated from a magpie (2021.TE.11). The replicon plasmid repB (plasmid pK2044) was detected in only one isolate. 

No plasmids were found in 7 *K. pneumoniae* isolates isolated from fallow deer, roe deer, deer, and one European badger.

### 3.5. cgMLST Analysis

Cluster analysis based on allele calling was performed on 16 out 17 isolates confirmed as *K. pneumoniae*.

Results showed that *K. pneumoniae* isolates clustered into sublineages concordant with MLST (Figure 3). The pair 2021.TE.19 and 2021.TE.20, isolated from wild boar and belonging to the same ST (3071), showed identical cgMLST profiles (cgST) according to Hennart et al. (2021). Furthermore, the pair 2021.TE.3 and 2021.TE.4, from fallow deer, presented the same cgST. These pairs may result from recent transmission.

The two genomes detected from wild boar (2021.TE.17 and 2021.TE.18) were uploaded in the Pathogenwatch platform in order to identify similar available clinical genomes. In particular, those two isolates were chosen because (i) 2021.TE.17 is shown to be hypervirulent and to belong to a human-related sequence typing (ST23) and (ii) 2021.TE.18 (ST35) displayed several AMR genomic features harboring tetracycline, sulfonamides, and aminoglycosides resistance genes.

The cluster analysis performed with 45 available clinical genomes of ST23 from Pathogenwatch showed that 2021.TE.17 was seven alleles distant from two isolates detected in a French horse in 1985 (Figure 4a).

Seven out of 45 genomes (ERR1761497, SRR9208902, ERR1218754, SRR9208903, SRR9208900, SRR9208898, and SRR9208899) harbored carbapenemase resistance genes.

The other isolate (2021.TE.18) was nine alleles distant from an Italian clinical strain (SRR10058597) isolated in 2016 (Figure 4b).

## 4. Discussions

Our results indicate that overall, the occurrence of *K. pneumoniae* in wild animals in Central Italy is low (12.2%). The phenotypic and genomic characterization of AMR in 16 *K. pneumoniae* and one *K. quasipneumoniae* isolates from wild animals allow us to deepen knowledge on this pathogen in this ecological niche. 

The antimicrobial susceptibility testing results highlighted significant levels of phenotypic resistance against β-lactams (ampicillin, cloxacillin and cefazolin), ertapenem, tetracyclines, and trimethoprim. Furthermore, we observed the presence of nine MDR *K. pneumoniae* isolates from a wolf, a deer, a magpie, a European badger, two fallow deer, and three wild boars. Moreover, investigating all the resistance genes detected in the dataset, *bla*_SHV_ genes were observed in all the 16 *K. pneumoniae* isolates and *bla*_OKP-A-2_ in only the *K. quasipneumoniae* isolate, as previously described by Wyres et al. [40]. Only one isolate harbored other resistance genes such as *sul*2, *tet*A, *str*A, and *str*B, which are responsible for resistance to sulfonamides, tetracycline, and aminoglycosides, respectively, although full consistency with the phenotypic profile was not observed.

The widespread use of tetracycline in the veterinary sector and agriculture for its broad spectrum of activity has increased the bacterial community’s tolerance to this compound [41], even if only one isolate detected in a wild boar harbored the *tet*A gene. 

On carbapenems, more than half of the isolates were resistant to ertapenem, even if none of the carbapenemase genes were found. The presence of resistant isolates in our study could be linked to the loss or deficiency of outer membrane porins (OMPs) that play an important role in conferring carbapenem resistance in *K. pneumoniae* [40]. Carbapenems are last resort drugs used in clinical settings, and although the prevalence of carbapenem-resistant *Enterobacteriaceae* (CRE) is low in animals, it is increasing and has been reported in companion animals, livestock, and even in wildlife and environmental samples [2]. The unusually high resistance rate to this compound may be explained by the increasing contact between wild animals and human-related environments, such as urban, agricultural, or livestock wastewaters, which are known to be some of the major reservoir of antibiotics and resistant bacteria [5].

Many available studies have been carried out on other *Enterobacteriaceae*, in particular *Escherichia coli*, and have shown high antibiotic resistance rates from wild animals in other Italian regions [8,9,10,11]. In Italy, few data are available on *K. pneumoniae* occurrence in wild animals and its characterization of AMR, and the data concerns wild birds only [42,43]. The available studies in Europe concern wild boars from Algeria [44], Chinese hares [45], hedgehogs in Spain [46], and European mouflons in Austria and Germany [47].

On mobile genetic elements, both the incompatibility group (Inc) plasmids and the colicin (Col)-type plasmid were detected in most of the isolates. Additionally, the replicon plasmid pK2044 was detected in the hypervirulent ST23 isolate from a wild boar, representing a possible concern. Indeed, the plasmid pK2044 was first described in 2009 by Wu et al. [48] and was found in an invasive *K. pneumoniae* isolate that caused liver abscesses and meningitis in humans. 

Recently, the emergence from clinical specimens of a hypervirulent ST23 clone carrying carbapenemase genes has subverted the “classical” division among hypervirulent and AMR clones [49]. Moreover, the convergence of antimicrobial resistance and virulence in hypervirulent isolates represents a public health concern due to the onset of difficult-to-treat infections in previously healthy adults [49].

As previously mentioned, the hypervirulent ST23 strain isolated from wild boar was shown to be similar to clinical isolates carrying carbapenemase genes, as revealed by the cgMLST analysis. In this study, the similarity among *K. pneumoniae* isolates isolated from a wild boar and clinical cases was also demonstrated for the MDR ST35 clone. These results are concordant with other authors [5,50,51], which show that identical or near identical strains belonging to the same clonal complex could circulate in wildlife, humans, and domestic animals.

In this study, isolates were mainly detected from the intestinal contents, confirming the carrier function of animals. Although the necropsies did not reveal any lesions, four isolates were detected in the brain and we cannot exclude the pathogenic potential of *K. pneumoniae*. Furthermore, some of these wild animals, such as wild boar, are also food-producing animals (e.g., Italy), where, according to local tradition, it is customary to eat seasoned raw meat products (e.g., salami, sausages). These traditional foods represent a possible way of transmission to humans, as has been demonstrated for other pathogens such as *Brucella* spp., and *Mycobacterium* spp. [52].

The zoonotic potential of *K. pneumoniae* in food-producing animals was widely reviewed by Davis and Price [53] and recently by Rodrigues et al. [54]. They described the contamination of milk and a variety of retail meats including poultry, beef, and pork, as well as freshwater fish and prawns, by *K. pneumoniae*. Furthermore, Davis et al. [55] and Rodrigues et al. [54] described the phylogenetic relatedness between urinary tract infection isolates with retail meat isolates sampled in the same geographic area.

In this work, various species of wild animals characterized by distinct ecological contexts have also been considered. According with Ramey & Ahlstrom [56], these species, as wild boars, can be defined as peridomestic animals, because they reliant upon human-dominated activities or managed by human habitats. The wildlife can be exposed to anthropogenic antibiotic resistance contamination through agricultural activities, livestock production practices, waste disposal, and wastewater management.

The overlapping of wildlife habits with livestock and human ones highlight that wild animals can represent a potential reservoir of AMR bacteria and can reveal the impact of human activities in the environment [1,52].

For this, in-depth knowledge about the ecological context of wildlife can provide insight into the environmental pathways through which antimicrobial determinants can be lost or acquired [57].

## 5. Conclusions

Despite the renewal of interest in *K. pneumoniae* epidemiology, there is a critical lack of an unbiased ecological approaches to defining the reservoirs and sources of infections. Most studies on risk factors of *K. pneumoniae* colonization or infection have focused on clinically-oriented questions, such as detection of ESBL-producing *K. pneumoniae*. The current knowledge of the ecological context of *K. pneumoniae* is still lacking, and the potential health risks are underestimated. 

This study highlighted the link between the interface of humans, animals, and environment, showing once more the importance of establishing collaboration strategies between public health actors to control and prevent the spread of AMR bacteria.

For these reasons, continuous data collections in molecular epidemiology and resistance development in wildlife, environment, and foods must be implemented.

## Figures and Tables

**Figure 1 animals-12-01347-f001:**
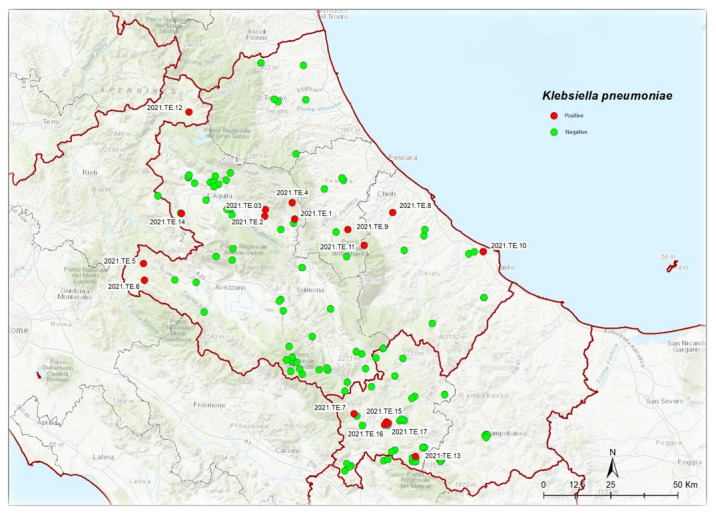
*K. pneumoniae* positive (red) and negative (green) samples isolated from wild animals in the Abruzzo, Molise, and Lazio regions.

**Figure 2 animals-12-01347-f002:**
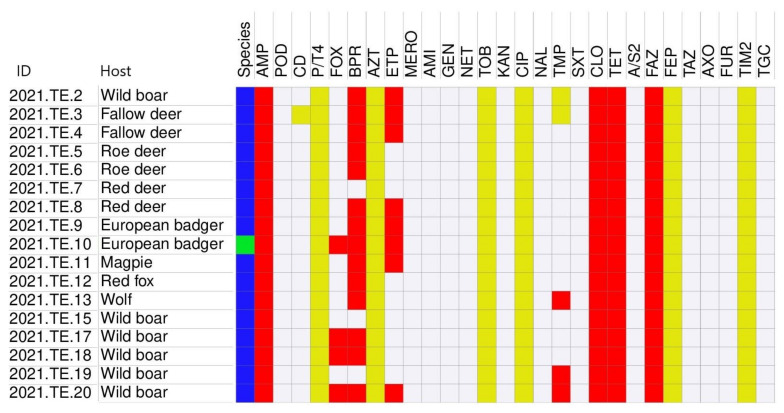
Heatmap encompassing antimicrobial susceptibility testing results of *K. pneumoniae* strains (blue) and *K. quasipneumionae* strain (green). Red color indicates resistance, yellow indicates an intermediate phenotype, and white indicates susceptibility. AMP: ampicillin, POD: cefpodoxime, CD: cefpodoxime-clavulanic acid, P/T4:piperacillin-tazobactam constant 4, FOX: cefoxitin, BPR: ceftobiprole, AZT: aztreonam, ETP: ertapenem, MERO: meropenem, AMI: amikacin, GEN: gentamicin, NET: netilmicin, TOB: tobramycin, KAN: kanamycin, CIP: ciprofloxacin, NAL: nalidixic acid, TMP: trimethoprim, SXT: trimethoprim-sulfamethoxazole, CLO: cloxacillin, TET: tetracycline, A/S2: ampicillin-sulbactam, FAZ: cefazolin, FEP: cefepime, TAZ: ceftazidime, AXO: ceftriaxone, FUR: cefuroxime, TIM2: ticarcillin-clavulanic acid constant 2, TGC: tigecycline.

**Figure 3 animals-12-01347-f003:**
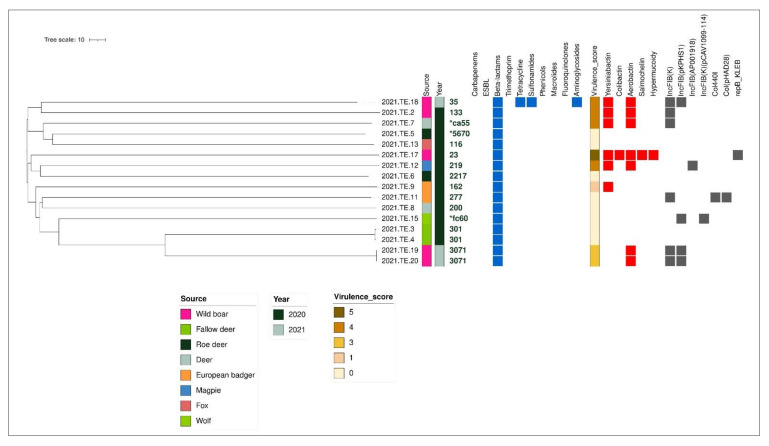
Neighbor joining (NJ) tree resulting from the cgMLST analysis performed on the 16 *K. pneumoniae* draft assemblies according to the Pasteur Institute’s scheme (629 loci). The tree is annotated with metadata (source and year of isolation), antimicrobial resistance gene (blue color indicates the presence), virulence score, virulence genes (red color indicates the presence), and plasmid (grey color indicates the presence). For novel sequence types we attributed the symbol “*”.

**Figure 4 animals-12-01347-f004:**
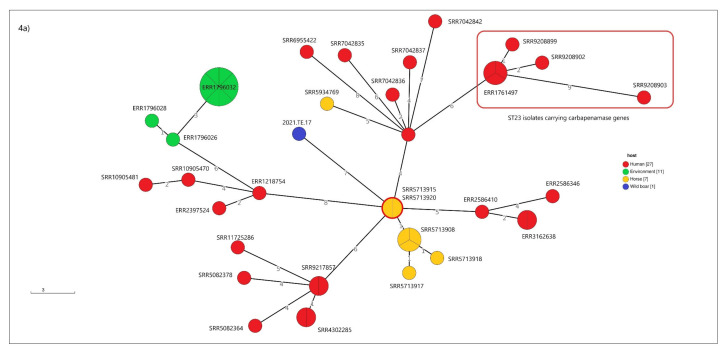
Cluster analysis performed with public genomes available on Pathogenwatch platform. (**a**) Core-genome MLST (cgMLST) analysis of the isolate 2021.TE.17 with 45 available clinical genomes belonging to the ST23. In the red circle are highlighted the ST23 public genomes carrying carbapenemase resistance genes. (**b**) cgMLST analysis of isolate 2021.TE.18 with four clinical genomes (ERR1217151, SRR5386627, SRR10058597, and ERR1541571). In the branches it is reported the numbers of allelic differences.

**Table 1 animals-12-01347-t001:** Metadata of 17 strains used in this study.

ID	Year	Host	Matrix
2021.TE.2	2020	Wild boar	Feces
2021.TE.3	2020	Fallow deer	Feces
2021.TE.4	2020	Fallow deer	Feces
2021.TE.5	2020	Roe deer	Feces
2021.TE.6	2020	Roe deer	Feces
2021.TE.7	2020	Red deer	Feces
2021.TE.8	2020	Red deer	Feces
2021.TE.9	2020	European badger	Small intestine
2021.TE.10	2020	European badger	Feces
2021.TE.11	2020	Magpie	Small intestine
2021.TE.12	2020	Red fox	Small intestine
2021.TE.13	2020	Wolf	Feces
2021.TE.15	2020	Wild boar	Brain
2021.TE.17	2020	Wild boar	Small intestine
2021.TE.18	2021	Wild boar	Brain
2021.TE.19	2021	Wild boar	Brain
2021.TE.20	2021	Wild boar	Brain

**Table 2 animals-12-01347-t002:** Genomic characterization of 17 *Klebsiella* spp. strains analyzed in this study. ST: sequence type, KL: K locus, O: O locus, β-lactams: beta-lactams, Tet: tetracycline, Sul: sulfonamides, Ami: aminoglycosides, Ybt: yersiniabactin, Clb: colibactin, Iuc: aerobactin, Iro: salmochelin, rmpA: hypermucoid genes.

ID	ST	KL	O	β-Lactams	*Tet*	*Sul*	Ami	Plasmid	Ybt	Clb	Iuc	Iro	rmpA
2021.TE.2	133	KL116	O1v1	SHV-75	/	/	/	IncFIB(K)	ybt 9 ICEKp3	/	unknown	/	/
2021.TE.3	301	KL116	O2v1	SHV-27	/	/	/	/	/	/	/	/	/
2021.TE.4	301	KL116	O2v1	SHV-27	/	/	/	/	/	/	/	/	/
2021.TE.5	*5670	KL30	O3/O3a	SHV-11	/	/	/	/	/	/	/	/	/
2021.TE.6	2217	KL13	O3b	SHV-1	/	/	/	/	/	/	/	/	/
2021.TE.7	*ca55	KL30	O1v1	SHV-65	/	/	/	IncFIB(K)	ybt 9 ICEKp3	/	iuc 3	/	/
2021.TE.8	200	*KL13	O3b	SHV-1	/	/	/	/	/	/	/	/	/
2021.TE.9	162	KL13	O3b	SHV-1	/	/	/	/	ybt 16 ICEKp12	/	/	/	/
2021.TE.10	4895	KL35	O3/O3a	OKP-A-2	/	/	/	IncFIB(K)	/	/	/	/	/
2021.TE.11	277	/	O3b	SHV-27	/	/	/	IncFIB(K), Col(pHAD28)	/	/	/	/	/
2021.TE.12	219	KL121	O1v1	SHV-1	/	/	/	IncFIB(AP001918)	ybt 16 ICEKp12	/	iuc 3	/	/
2021.TE.13	116	KL11	O3/O3a	SHV-1	/	/	/	/	/	/	/	/	/
2021.TE.15	*fc60	KL117	O2v2	SHV-11	/	/	/	IncFIB(pKPHS1) IncFIB(K)(pCAV1099-114)	/	/	/	/	/
2021.TE.17	23	KL1	O1v2	SHV-11	/	/	/	repB (pK2044)	ybt 1 ICEKp1	clb 2	iuc 1	iro1	rmpA1 rmpA2
2021.TE.18	35	KL22	O1v1	SHV-33	tetA	sul2	strA strB	IncFIB(pKPHS1) IncFIB(K)	ybt 5 ICEKp6	/	iuc 3	/	/
2021.TE.19	3071	KL31	OL104	SHV-27	/	/	/	IncFIB(pKPHS1) IncFIB(K)	/	/	iuc 3	/	/
2021.TE.20	3071	KL31	O3b	SHV-27	/	/	/	IncFIB(pKPHS1) IncFIB(K)	/		iuc 3	/	/

## Data Availability

The whole genome projects have been deposited in the Nation Center for Biotechnology Information (NCBI) under the BioProject accession number PRJNA774508.

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
