# Peer review of "Phenotypic and Genetic Characterization of Klebsiella pneumoniae Isolates from Wild Animals in Central Italy"

_animals, 2022, doi:10.3390/ani12111347_

Round 1

Reviewer 1 Report

Dear Authors,

please find below my comments on your paper.

I enjoyed it and I learned also a lot while reading, therefore, I would like to see it published.

IN GENERAL

In my opinion, this paper has the following strenghts:

  • It is well written
  • AMR is an important topic deserving growing attention
  • The insight on the pneumoniae isolates is deep
  • The results are per se interesting

Notwithstanding, I see the following weaknessess:

  • Authors stress too much the eco-epidemiological significance of their data. Actually, the sample is too low, too scattered on a wide area and composed by too many host species with different ecology to be representative in an eco-epidemiological perspective.
  • The concept of “reservoir”, in my opinion, is misleading in this paper. Actually, for “reservoir of AMR pneumoniae” I would expect a species (or, better, a population) within which K. pneumoniae infection cycle is mantained, as well as the features of AMResistance. Of course it cannot be excluded that this is the case, but it should be kept in mind that being susceptible to a pathogen’s infection does not automatically imply to become a reservoir. Moreover, in case of infection by a AMR bacterial strain in a wild population, do the Authors think that AMResistance would persist or could be lost in time? I guess for sure it depends on antibiotic pressure, but I think it cannot be homogeneous in such a big area as the one considered in the paper. For the reasons above, maybe the term “sentinel” is preferable.

This issues make the discussion and conclusions of the paper not at the same height of its results.  I suggest no to use these results for speculative ecological discussion. Rather, based on these results, I would discuss about what would be needed to investigate the eco-epidemiology of AMR K. pneumoniae. As an example, selecting specific host species (and sympatric livestock/pets), locations “at risk”…

SOME SUGGESTIONS:

  • Some animals are reported to have died for gunshot wounds. Please specify whether they were regularly hunted, or poached, or other
  • When using MIC method, is still worthwhile to use also disk diffusion method?
  • I would use the term “source” for wastewater and other fomites instead of “reservoir”
  • African Swine Fever is not a zoonosis and cannot be transmitted to humans

Sincerely

Author Response

Thank you for the improving suggestions. Please find the attached file for seeing the related comments and/or modifications.

Reviewer 2 Report

The characterization of Klebsiella isolates from wildlife is really interesting for the epidemiology of this nosocomial pathogen. In this study, isolates have been analyzed focusing not only on phenotypic AMR but also on antimicrobial resistance genes (ARG). The study of ARG on the ecosystems is becoming more and more important to understand the scope of the greatest public health issue of the XXI century. 

First of all, I would like to congratulate the authors for the study. It is really complete and the results are very interesting. 

However, I have few minor recommendations to improve the manuscript: 

Simple Summary and Abstract: 

Line 17, 26 and 37: instead WGS, it would be more grammatically correct to write the complete name of the technique (whole genome sequence). I understand that you have a word limit in the simple summary and abstract, but at least in keywords... 

Line 30: the same situation with MDR (multridrug resistance I think). 

Introduction: 

Line 46: authors should add a reference to highlight the importance of the link between human, livestock, and natural environments. For example: 

Martín-Maldonado, B., Montoro-Dasi, L., Pérez-Gracia, M. T., Jordá, J., Vega, S., Marco-Jiménez, F., & Marin, C. (2019). Wild Bonelli’s eagles (Aquila fasciata) as carrier of antimicrobial resistant Salmonella and Campylobacter in Eastern Spain. Comparative Immunology, Microbiology and Infectious Diseases67, 101372.

Line 59: Western world is too colloquial for me. Maybe it could be replaced, but it is only a personal opinion. 

Line 64: replace "the formation of multidrug resistant" for "the development of multidrug resistant". 

Material and Methods: 

I have some questions about the sample collection. I understand that the animals included in the study depend on several surveillance programs. You collected a total of 131 samples from 119 animals, but for some animals, the samples were feces, for others, intestine or brain. I think it is important to justify why or in which animals you collected one or other samples. Also, it will be interesting to know an average of how much time has passed between the death of the animals and the sample collection, as the microbiome can change after death. 

Information about the sample's preservation until the analysis is missing (if they are stored in refrigeration, in a specific medium, and how much time has been stored in that conditions until their analysis at the lab).  

Line 123: room temperature should be more specific... it was 15ºC, 25ºC, 40ºC... just add an approximation in brackets. 

Line 128: Escherichia coli should be in italics. 

Line 152 and 157: the GN3F plate specifications should be in line 152 as it is the first time you mentioned it. 

Results: 

The order of Material and Methods should be the same of the Results. In the first headland, you described the isolation and characterization, followed by phenotypic AMR, WGS, and gene profiles. In the Results, you described first the isolation and characterization, followed by WGS results before the AMR results. I recommend you order the results... 

Lines 227 and 228: I think the description of species from which the isolates were obtained is not necessary as it is repeated in table 1 (and it is more visual in tabular form). 

Line 255: Before the description of antimicrobial resistance test results, it is important to know the percentage of isolates resistant to at least one antimicrobial. You mentioned it in the abstract, but you didn't in Results. Please add the percentage. 

Line 260: " (18.8%) isolated from wild boars". It is not necessary to describe from which species have been isolated the resistant strains. With the information in Table 1 and in Figure 2, it is easy to figure out. Moreover, if you want to simplify the lecture, just adding a column in Figure 2 with the species would be enough. The same situation is in Line 263, 285, 314, and 358. 

Line 262: replace "and" for "but". 

Lines 277 to 283: simplify please, it is very repetitive. 

Line 285: replace "and" for "but". 

Line 286: information in brackets is included in Material and Methods, delete it in this line, please. 

Line 288: change "Moreover" for "Finally". 

Discussions: 

Klebsiella has an intermittent excretion in faces, so the detection could be underestimated in animals in which the sample collected was feces. It would be interesting to discuss this point...

Line 368: a reference about the use of tetracycline in veterinary and agriculture is missing. For example: 

Santás-Miguel, V., Rodríguez-González, L., Núñez-Delgado, A., Álvarez-Rodríguez, E., Díaz-Raviña, M., Arias-Estévez, M., & Fernández-Calviño, D. (2022). Time-course evolution of bacterial community tolerance to tetracycline antibiotics in agricultural soils: A laboratory experiment. Chemosphere291, 132758.

Palma, E., Tilocca, B., & Roncada, P. (2020). Antimicrobial resistance in veterinary medicine: An overview. International Journal of Molecular Sciences21(6), 1914.

Line 396: a reference about difficult-to-treat infections and the public health concern for AMR is missing. For example: 

La Rosa, R., Johansen, H. K., & Molin, S. (2022). Persistent Bacterial Infections, Antibiotic Treatment Failure, and Microbial Adaptive Evolution. Antibiotics11(3), 419.

Line 406-407: reword

Author Response

(The authors gave the same response as above.)
